# Anemia among HIV-positive women in LMICs: Multilevel analysis of recent DHS survey

Enyew Getaneh Mekonen *

Department of Surgical Nursing, School of Nursing, College of Medicine and Health Sciences, University of Gondar, Gondar, Ethiopia

* enyewgetaneh12@gmail.com

## Abstract

### Introduction

Anemia affects over 30% of women of reproductive age globally, with the highest burden in low- and middle-income countries, and it poses additional risks for women living with human immunodeficiency virus (HIV), including disease progression and reduced survival. Although previous studies report prevalence among HIV-positive women ranging from 37.8% to 55.8%, most evidence comes from hospital-based or high-income settings, leaving a gap in population-level data. Addressing this gap is critical, as women in low- and middle-income countries (LMICs) often face overlapping vulnerabilities such as nutritional deficiencies, limited healthcare access, and high HIV burden. Using nationally representative Demographic and Health Surveys, this study aims to estimate anemia prevalence and identify associated factors among HIV-positive women to inform targeted interventions and integrated management strategies.

### Methods

A cross-sectional study was conducted using Demographic and Health Survey data collected between 2022 and 2024 from nine countries in sub-Saharan Africa and Asia, including 1,446 HIV-positive women aged 15–49 years. Hemoglobin concentration was used to classify anemia based on World Health Organization (WHO) thresholds. Individual and community-level factors were examined, and weighted data were analyzed using multilevel logistic regression to account for clustering. Associations were reported as adjusted odds ratios with 95% confidence intervals.

### Results

Among HIV-positive women in Africa and Asia, the prevalence of anemia was 50.62% (95% confidence interval (CI): 48.04–53.20%), with 19.29% classified as mild, 26.28% as moderate, and 5.05% as severe. Prevalence varied widely

**Data availability statement:** The data is publicly available online at https://dhsprogram.com/data/available-datasets.cfm.

**Funding:** The author(s) received no specific funding for this work.

across countries, ranging from 71.43% in Mali to 12.00% in Tajikistan. Educational status [adjusted odds ration (AOR) = 0.43; 95% CI: 0.22–0.81], media exposure [AOR = 0.41; 95% CI: 0.19–0.87], contraceptive use [AOR = 2.37; 95% CI: 1.35–4.17], and iron supplementation during pregnancy [AOR = 2.17; 95% CI: 1.04–4.55] were significantly associated with anemia.

## Conclusions

Anemia remains a major public health concern among HIV-positive women, driven by reproductive, nutritional, and socio-behavioral factors. Strengthening antenatal and HIV care programs, integrating family planning services, and promoting adherence to iron supplementation are critical strategies to reduce anemia risk. Tailored health communication and nutritional interventions, alongside future longitudinal studies, are essential to establish causal pathways and inform targeted interventions.

## 1. Introduction

Anemia remains a major public health concern globally, affecting over 30% of women of reproductive age, with the highest burden observed in low- and middle-income countries (LMICs) [1]. It is linked to increased morbidity and all-cause mortality, decreased economic productivity, and diminished cognitive and physical abilities [2–4]. Anemia during pregnancy has been proposed as a potential marker of increased risk of major hemorrhage and a risk factor for maternal death [5,6]. Maternal iron deficiency can result in unfavorable pregnancy and neonatal outcomes, such as stillbirth, low birth weight, and infant mortality [7,8].

People living with the human immunodeficiency virus (HIV) may experience severe consequences from anemia, ranging from a decline in functioning and quality of life to a correlation with the advancement of the disease and a lower chance of survival [9]. In LMICs, where both HIV prevalence and anemia rates are disproportionately high, women face compounded vulnerabilities [10–13]. Limited access to healthcare, poverty, gender inequality, and stigma further exacerbate the risk and impact of anemia among HIV-positive women [9]. Fatigue, dyspnea, and elevated heart rate are all consequences of anemia that significantly impair a patient's quality of life [14].

Anemia in patients with HIV is caused by a variety of factors. Hematopoietic stem/progenitor cells (HSPCs) in the bone marrow may be affected by HIV both directly and indirectly [15,16]. The proliferation and differentiation of HSPCs during hematopoiesis may potentially be impacted by medications used for antiretroviral therapy (ART), inflammatory mediators generated during HIV infection, coinfections, or opportunistic infections [15,16]. This may cause hematologic disorders like thrombocytopenia that may be linked to a number of bleeding abnormalities [17], which may raise the risk of anemia or potentially make it worse [18].

Studies conducted elsewhere showed that the prevalence of anemia among HIV-positive women ranges from 37.8% to 55.8% [10–13]. Factors like educational status, contraceptive use, pregnancy status, breastfeeding status, health

insurance coverage, residence, iron supplementation, body mass index, sex of household head, type of toilet facilities, and menstruation within six weeks prior to data collection were significantly associated with anemia prevalence [11,13,19].

Although anemia among HIV-positive women has been widely studied, most existing research has focused on hospital-based populations or cohorts in high-income countries, limiting its generalizability to LMICs. This gap is critical, as women in LMICs often face overlapping vulnerabilities, including nutritional deficiencies, limited healthcare access, and high HIV burden. To address this gap, recent nationally representative Demographic and Health Survey (DHS) datasets were analyzed from nine LMICs in sub-Saharan Africa and Asia. By leveraging population-level data across diverse community settings, this study provides novel evidence on anemia prevalence and risk factors among HIV-positive women outside of hospital contexts, offering cross-country insights that can inform targeted public health interventions in resource-limited settings.

The DHS provide nationally representative data on key health indicators, including hemoglobin concentration and HIV status. The DHS dataset offers a unique opportunity to examine anemia prevalence and its associated factors among HIV-positive women across multiple countries using standardized methodology [20]. Hemoglobin levels are measured using portable HemoCue analyzers and recorded as variable v456, while HIV status is captured in v781. These variables allow for robust analysis of anemia prevalence and its sociodemographic, behavioral, and information-related correlates. This study aims to estimate the prevalence of anemia and identify associated factors among HIV-positive women in LMICs using the most recent DHS data.

## 2. Methodology

### 2.1. Study design

A cross-sectional study using secondary data from DHS surveys conducted between 2022 and 2024 was employed.

### 2.2. Data sources, sampling, and populations

Recent DHS datasets from selected countries in sub-Saharan Africa (SSA) (Democratic Republic of Congo, Ghana, Lesotho, Mali, Mozambique, Nigeria, and Tanzania) and Asia (Nepal and Tajikistan) were employed. Every five years, the community-based cross-sectional DHS study is carried out to generate updated demographic and health-related data. Women aged 15–49 years who participated in the DHS surveys and who were residents of each country's randomly selected enumeration areas during the survey year were eligible for inclusion. All women who tested positive for HIV in the DHS datasets were included, regardless of antiretroviral therapy (ART) status, pregnancy, or comorbid conditions, as DHS does not routinely collect detailed clinical data on these variables. Women were excluded if they had missing or incomplete data on hemoglobin measurements, HIV test results, or key sociodemographic covariates required for analysis. To determine the prevalence of anemia and factors associated with it among HIV-positive women aged 15–49 years in nine SSA and Asian countries, the data were appended. Different datasets, such as those for children, men, women, births, and households, are included in the survey for each nation. The individual's record (IR file) was employed in the current investigation. The DHS is a national survey that is primarily conducted in LMICs every five years. To enable cross-country comparison, consistent techniques are used for sampling, questionnaires, data collection, cleaning, coding, and analysis [21]. The study included a weighted sample of 1,446 HIV-positive women who fully responded to all factors of interest (Table 1). The DHS uses a two-stage, stratified sampling method [22]. The first step is creating a sample frame, which is a list of enumeration areas (EAs) or primary sampling units (PSUs) that encompass the entire nation. This list is typically created using the most recent national census that is available. The systematic sampling of the homes included in each cluster, or EA, is the second step. More details on survey sample techniques are available in the DHS guidelines [23].

**Table 1. Sample size for anemia and its associated factors among HIV-positive women in Africa and Asian countries.**

| Country | Year of survey | Weighted sample (n) | Weighted sample (%) |
|---|---|---|---|
| Democratic Republic Congo | 2023–24 | 17 | 1.17 |
| Ghana | 2022 | 12 | 0.83 |
| Lesotho | 2023–24 | 642 | 44.40 |
| Mali | 2023–24 | 7 | 0.48 |
| Mozambique | 2022–23 | 464 | 32.09 |
| Nepal | 2022 | 1 | 0.07 |
| Nigeria | 2023–24 | 53 | 3.67 |
| Tajikistan | 2023 | 25 | 1.73 |
| Tanzania | 2022 | 225 | 15.56 |
| **Total sample size** | | 1,446 | 100 |

### 2.3. Variables of the study

**2.3.1. Outcome variable.** Hemoglobin concentration is measured in grams per deciliter (g/dl) and recorded in DHS as variable v456, with values multiplied by 10 (i.e., 120 = 12.0 g/dl). Anemia status is defined as follows:

**1. Any Anemia**

A woman is classified as having anemia if:
She is non-pregnant and has hemoglobin < 12.0 g/dl (v456 < 120), or
She is pregnant and has hemoglobin < 11.0 g/dl (v456 < 110).

**2. Mild Anemia**

A woman is classified as having mild anemia if:
She is non-pregnant and has hemoglobin between 11.0 and 11.9 g/dl (v456 in 110:119), or
She is pregnant and has hemoglobin between 10.0 and 10.9 g/dl (v456 in 100:109).

**3. Moderate Anemia**

A woman is classified as having moderate anemia if:
She is non-pregnant and has hemoglobin between 8.0 and 10.9 g/dl (v456 in 80:109), or
She is pregnant and has hemoglobin between 7.0 and 9.9 g/dl (v456 in 70:99).

**4. Severe anemia**

A woman is classified as having severe anemia if:
She is non-pregnant and has hemoglobin less than 8.0 g/dl (v456 < 80).
She is pregnant and has hemoglobin less than 7.0 g/dl (v456 < 70).

**2.3.2. Independent variables.** Both individual- and community-level variables were considered to accommodate the hierarchical nature of DHS data.

**2.3.3. Individual-level variables.** Variables included at this level were respondent's age (15–24 years, 25–34 years, 35–49 years), educational status (no education, primary, secondary/higher), marital status (unmarried, married), working status (not working, working), wealth index (poor, middle, rich), media exposure (no, yes), currently pregnant (no, yes), currently breastfeeding (no, yes), ever had terminated pregnancy (no, yes), cigarette smoking (no, yes), body mass index (underweight, normal, overweight, obese), number of children (no child, 1–2 children, > 2 children), distance to

health facility (big problem, not a big problem), type of toilet facility (unimproved, improved), source of drinking water (unimproved, improved), modern contraceptive utilization (no, yes), sex of the household head (male, female), health insurance coverage (no, yes), menstruation within six weeks (no, yes), and iron supplementation (no, yes).

**2.3.4. Community-level variables.** Variables considered at the community level were place of residence (urban, rural), community literacy level (low, high), community media exposure (low, high), and community poverty level (low, high).

**2.3.5. Description of independent variables. Media exposure**: Individual media exposure was determined by combining three indicators: reading newspapers, watching television, and listening to the radio. Women were considered exposed to media if they engaged with at least one of these sources.

**Wealth index**: Categorized into three groups based on the DHS wealth index: poor (poorest and poorer), middle, and rich (richer and richest).

**Body Mass Index (BMI)**: Categorized according to WHO standards. Individuals with a BMI less than 18.5 kg/m² are considered underweight. Those with a BMI between 18.5 and 24.9 kg/m² are classified as having a normal weight. A BMI between 25.0 and 29.9 kg/m² is categorized as overweight, while a BMI of 30.0 kg/m² or higher is classified as obese [24].

**Type of toilet facility (unimproved or improved)**: Improved: flush or pour-flush to piped sewer system, septic tank, or pit latrine; ventilated improved pit latrine; pit latrine with slab; and composting toilet. Unimproved: flush or pour-flush to elsewhere, pit latrine without slab or open pit, bucket, hanging toilet or hanging latrine, and no facilities or bush or field (open defecation) [25].

**Source of drinking water (unimproved or improved)**: Improved: use of piped water into dwelling, piped water to yard/plot, public tap/standpipe, tube well or borehole, protected well, protected spring, and rainwater collection; Unimproved: use of unprotected wells, unprotected springs, surface water (river, dam, lake/ponds/stream/canal/irrigation channel), tanker trucks, and carts with small tanks [25].

**Community literacy level**: the proportion of women with a minimum primary level of education derived from data on respondents' level of education. Then, it was categorized using the national median value into two categories: low (communities with ≤50% of women having at least primary education) and high (communities with >50% of women having at least primary education).

**Community media exposure**: Calculated as the proportion of women in each cluster who had exposure to at least one media source (television, radio, or newspapers). Communities were classified as having low media exposure (<50%) or high media exposure (≥50%).

**Community poverty level**: Derived from the percentage of women in the poorest and poorer wealth quintiles within each cluster. Communities were categorized as low poverty (<50%) or high poverty (≥50%).

## 2.4. Data management and analysis

Data from the most recent DHS survey were cleaned, recoded, and analyzed using STATA/SE version 17.0. Sample weights were applied to account for sampling errors and non-responses. Continuous variables were categorized, and categorical variables were further recoded as needed. Descriptive statistics, including frequencies, percentages, means, and standard deviations, were used to summarize individual and community-level variables in tables and figures. Due to the hierarchical nature of the DHS data, assumptions of independent observations and equal variance across clusters were violated, making traditional logistic regression unsuitable. Therefore, a multilevel logistic regression approach was adopted to assess factors associated with anemia.

Four models were constructed:

**Model 0 (Null Model)**: Included only the outcome variable to assess variability across clusters.

**Model I**: Included only individual-level predictors.

**Model II**: Included only community-level predictors.

**Model III**: Included both individual and community-level variables.

The intra-class correlation coefficient (ICC) and proportional change in variance (PCV) were calculated to evaluate the extent of clustering and the contribution of community-level factors to unexplained variance. The model with the lowest deviance was selected as the best fit. Variables with a p-value < 0.05 and adjusted odds ratios (AORs) with 95% confidence intervals (CIs) were considered statistically significant. Multicollinearity was assessed using the variance inflation factor (VIF), which ranged between 1 and 10, indicating no significant collinearity among predictors.

**2.4.1. Random effects.** Measures of variation and random effects for anemia were assessed using the ICC, median odds ratio (MOR), and PCV. Cluster-level variation was quantified through ICC and PCV, treating clusters as random effects. The ICC was calculated using the formula:

$$ICC = VC / (VC + 3.29) \times 100\%$$

Where VC represents the cluster-level variance, indicating the proportion of total variance attributable to differences between clusters.

The MOR reflects the median odds ratio comparing the likelihood of anemia between two randomly selected clusters, one with higher risk and one with lower risk, and was computed as

$$MOR = e^{0.95\sqrt{VC}}$$

The PCV estimates the proportion of variance explained by the inclusion of explanatory variables and was calculated as:

$$PCV = (V_{null} - VC)/V_{null} \times 100\%$$

Where $V_{null}$ is the variance from the null model and VC is the variance from the model with predictors [26]. Associations between anemia and both individual- and community-level variables were evaluated using fixed effects in the multilevel logistic regression models.

## 2.5. Ethical consideration

Permission was granted to download and use the data from https://dhsprogram.com/data/available-datasets.cfm before conducting the study. Ethical clearance was obtained from the Institutional Review Board of the DHS Program, ICF International. The procedures for DHS public-use data sets were approved by the Institutional Review Board. Identifiers for respondents, households, or sample communities were not allowed in any way, and the names of individuals or household addresses were not included in the data files. The number for each EA in the data file does not have labels to show their names or locations. There were no patients or members of the public involved since this study used a publicly available data set.

## 3. Results

### 3.1. Individual- and community-level characteristics of women

A total of 1,446 HIV-positive women were incorporated in the final analysis. The mean age of study subjects was 36.4±0.21 (SD), and 62.38% of them fell within the age range of 35–49 years. More than half (53.39%) of HIV-positive women were married, and 49.17% of them completed primary education. More than two-thirds (69.50%) of women had media exposure, and 60.37% of them had jobs. Only 3.46% of HIV-positive women were pregnant, and 47.72% of them had rich wealth status. The majority (88.73%) of respondents were not currently breastfeeding, and 17.84% of them have ever had a terminated pregnancy. Only 1.52% of study subjects smoke cigarettes, and 21.50% of them are obese.

More than half (57.13%) of HIV-positive women had two or more children, and 68.81% of them reported distance to a health facility was not a big problem. Among HIV-positive women, 50.55% used an unimproved type of toilet facility, while 72.89% relied on an improved source of drinking water. More than half (52.14%) of women reported not using modern contraceptives, and 51.11% lived in households headed by females. The majority (96.41%) of HIV-positive women were not covered by health insurance, and 68.95% of them had menstruation within six weeks. Among the study subjects, 83.02% reported receiving iron supplementation during their pregnancy, while 55.05% resided in rural areas. Among HIV-positive women, 70.95% lived in communities with high literacy levels, 55.60% in communities with high media exposure, and 56.43% in communities with low poverty levels (Table 2).

### 3.2. Prevalence of anemia among HIV-positive women

In the present study, 50.62% (95% CI: 48.04%−53.20%) HIV-positive women in the Africa and Asia regions were anemic. Of these, 19.29% of women had mild anemia, 26.28% had moderate anemia, and 5.05% had severe anemia (Fig 1).
   The prevalence of anemia was high in Mali (71.43%) and low in Tajikistan (12.00%) (Fig 2).

### 3.3. Measures of variation and model fitness

To evaluate the appropriateness of assessing random effects at the community level, a null model was first constructed. The results indicated significant variation in anemia across communities, with a cluster-level variance of 0.26 and a p-value < 0.001. Of the total variation in anemia, 7.43% was attributed to within-cluster differences, while 92.57% was due to variation between clusters. The MOR from the null model revealed that women in higher-risk clusters were 1.63 times more likely to have anemia compared to those in lower-risk clusters. Model I, which included only individual-level variables, yielded an ICC of 18.26%. Model II, incorporating only community-level variables, showed an ICC of 6.98%, further supporting the presence of substantial cluster-level variation. In the final model (Model III), which combined both individual and community-level variables, 5.15% of the variation in anemia was explained by these factors. The MOR in this model was 1.49, suggesting that the odds of anemia still varied significantly between clusters with low and high anemia rates, even after accounting for both levels of influence (Table 3).

### 3.4. Factors associated with anemia among HIV-positive women

In the final model (model III), factors like educational status, media exposure, modern contraceptive utilization, and iron supplementation during pregnancy were significantly associated with anemia. Accordingly, HIV-positive women who completed primary education were 57% less likely to develop anemia than those who completed secondary or higher education [AOR = 0.43; 95% CI (0.22, 0.81)]. Women who had no media exposure were 59% less likely to have anemia than their counterparts [AOR = 0.41; 95% CI (0.19, 0.87)]. The odds of anemia were 2.37 times higher among HIV-positive women who did not utilize modern contraceptives compared with those who utilized it [AOR = 2.37; 95% CI (1.35, 4.17)]. Non-adherence to iron supplementation during pregnancy increased the odds of developing anemia among HIV-positive women [AOR = 2.17; 95% CI (1.04, 4.55)] (Table 4).

## 4. Discussion

This study provides novel population-level evidence on anemia among HIV-positive women in LMICs, addressing a critical gap in the literature. By analyzing nationally representative DHS data from nine LMICs in sub-Saharan Africa and Asia, both the prevalence and key sociodemographic and health-related factors associated with anemia were identified. These findings contribute to a clearer understanding of context-specific risks and offer insights that can inform integrated management strategies within HIV care and women's health programs. Thus, 50.62% of HIV-positive women in the Africa and Asia regions were anemic. This prevalence is slightly lower than that reported in Nepal (55.8%) [12]. Several factors may explain this discrepancy. First, the Nepal study screened

**Table 2. Individual-and community-level characteristics of HIV-positive women aged 15-49 years in Africa and Asian countries, data from DHS 2022–2024.**

| Variables | Category | Frequency (n) | Percentage (%) |
|---|---|---|---|
| Respondent's age | 15-24 years | 132 | 9.13 |
| | 25-34 years | 412 | 28.49 |
| | 35-49 years | 902 | 62.38 |
| Educational status | No education | 163 | 11.27 |
| | Primary | 711 | 49.17 |
| | Secondary/Higher | 572 | 39.56 |
| Marital status | Unmarried | 674 | 46.61 |
| | Married | 772 | 53.39 |
| Working status | Not working | 573 | 39.63 |
| | Working | 873 | 60.37 |
| Media exposure | No | 441 | 30.50 |
| | Yes | 1,005 | 69.50 |
| Wealth index | Poor | 451 | 31.19 |
| | Middle | 305 | 21.09 |
| | Rich | 690 | 47.72 |
| Currently pregnant | No | 1,396 | 96.54 |
| | Yes | 50 | 3.46 |
| Currently breastfeeding | No | 1,283 | 88.73 |
| | Yes | 163 | 11.27 |
| Ever had terminated pregnancy | No | 1,188 | 82.16 |
| | Yes | 258 | 17.84 |
| Cigarette smoking | No | 1,424 | 98.48 |
| | Yes | 22 | 1.52 |
| Body mass index | Underweight | 65 | 4.50 |
| | Normal | 689 | 47.65 |
| | Overweight | 381 | 26.35 |
| | Obese | 311 | 21.50 |
| Number of children | No child | 116 | 8.02 |
| | 1-2 children | 504 | 34.85 |
| | >2 children | 826 | 57.13 |
| Distance to health facility | Big problem | 451 | 31.19 |
| | Not a big problem | 995 | 68.81 |
| Type of toilet facility | Unimproved | 731 | 50.55 |
| | Improved | 715 | 49.45 |
| Source of drinking water | Unimproved | 392 | 27.11 |
| | Improved | 1,054 | 72.89 |
| Modern contraceptive utilization | No | 754 | 52.14 |
| | Yes | 692 | 47.86 |
| Sex of the household head | Male | 707 | 48.89 |
| | Female | 739 | 51.11 |
| Health insurance coverage | No | 1,370 | 96.41 |
| | Yes | 51 | 3.59 |
| Menstruation within six weeks | No | 449 | 31.05 |
| | Yes | 997 | 68.95 |

*(Continued)*

**Table 2.** (Continued)

| Variables | Category | Frequency (n) | Percentage (%) |
|---|---|---|---|
| Iron supplementation | No | 63 | 16.98 |
| | Yes | 308 | 83.02 |
| Place of residence | Urban | 650 | 44.95 |
| | Rural | 796 | 55.05 |
| Community literacy level | Low | 420 | 29.05 |
| | High | 1,026 | 70.95 |
| Community media exposure | Low | 642 | 44.40 |
| | High | 804 | 55.60 |
| Community poverty level | Low | 816 | 56.43 |
| | High | 630 | 43.57 |

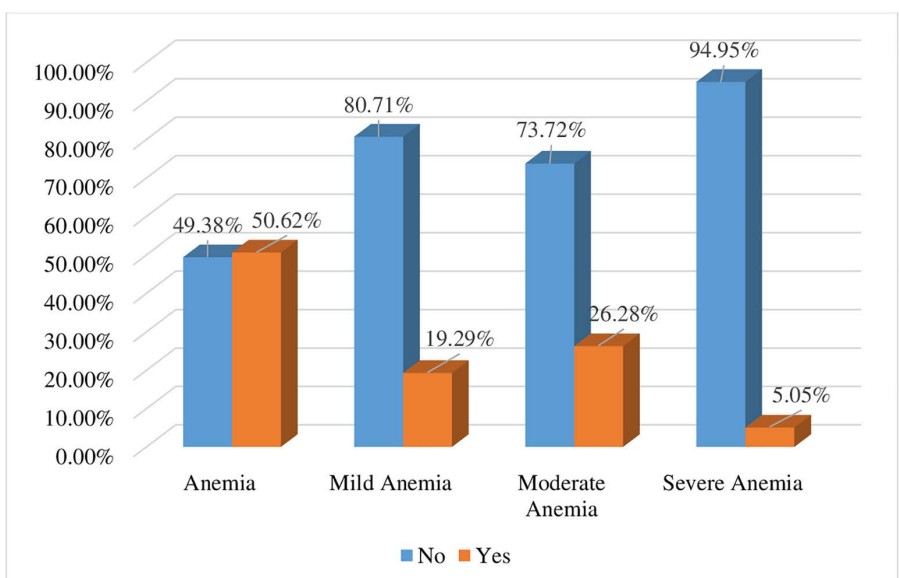

**Fig 1. Prevalence of anemia among HIV-positive women in Africa and Asia regions, data from DHS 2022–2024 (n = 1,446).**

a larger population, which may have increased the likelihood of detecting more cases, including severe anemia. Second, differences in survey years, nutritional environments, and HIV care coverage between Nepal and the countries included in this analysis may have influenced prevalence estimates. Third, variations in anemia case definitions and measurement approaches across studies could contribute to differences in reported prevalence. Taken together, these methodological and contextual differences provide a more robust explanation for why this study observed lower prevalence compared to the Nepal study, despite focusing exclusively on women. On the other hand, the prevalence observed in the current study is higher than reports from Ethiopia (23.9%) [19], Zimbabwe (37.8%) [11], and a pooled analysis across 18 sub-Saharan African countries (45.1%) [13]. This higher prevalence may reflect differences in study populations, nutritional status, antiretroviral therapy coverage, and methodological approaches across regions. This also reflects regional disparities in healthcare access, socioeconomic conditions, and reproductive health factors.

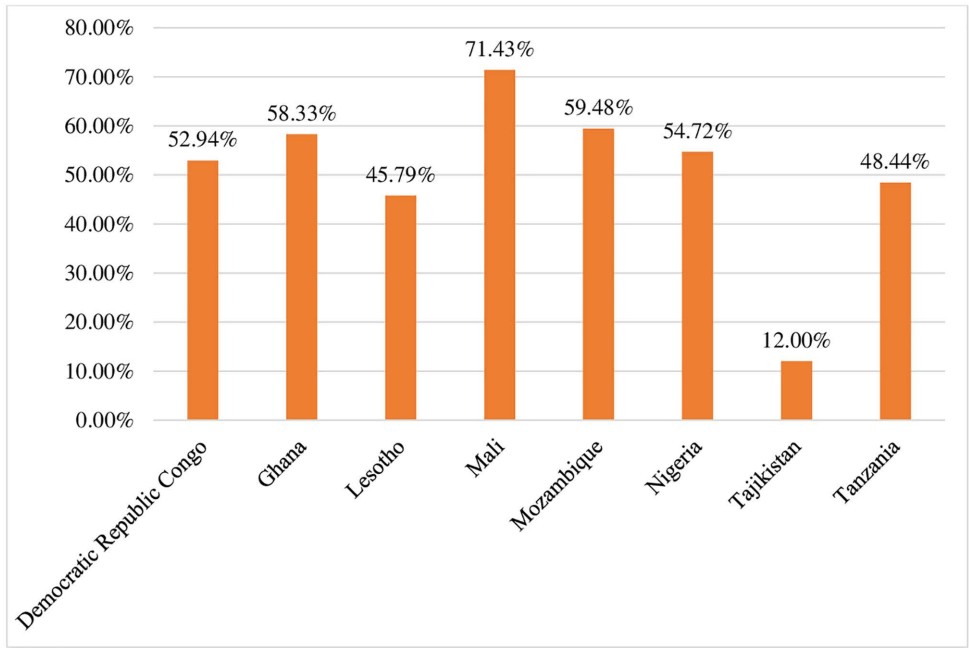

**Fig 2. Prevalence of anemia among HIV-positive women by country in Africa and Asia regions, data from DHS 2022–2024 (n = 1,446).**

**Table 3. Model comparison and random effect analysis for prevalence and associated factors of anemia among HIV-positive women aged 15-49 years in Africa and Asian countries.**

| Parameter | Null model | Model I | Model II | Model III |
|---|---|---|---|---|
| Variance | 0.2639906 | 0.2033494 | 0.2468102 | 0.1786759 |
| ICC | 7.43% | 5.82% | 6.98% | 5.15% |
| MOR | 1.63 | 1.54 | 1.60 | 1.49 |
| PCV | Reference | 22.97% | 6.51% | 8.53% |
| **Model fitness** | | | | |
| LLR | −999.39356 | −229.31549 | −997.26255 | −227.31687 |
| Deviance | 1,998.78712 | 458.63098 | 1,994.5251 | 454.63374 |

ICC: Intra cluster correlation, LLR: log-likelihood ratio, MOR: median odds ratio, PCV: Proportional change in variance.

This study also identified factors significantly associated with anemia. As a result, HIV-positive women who completed only primary education were less likely to develop anemia compared to those with secondary or higher education, suggesting that educational attainment may interact with socioeconomic, nutritional, or health-seeking factors in complex ways. This finding contradicts a study conducted in 18 SSA countries [13], which reported that higher educational attainment was generally protective against anemia among HIV-positive women. However, similar paradoxical associations have been reported elsewhere. For example, a study in Ghana found that urban women had higher anemia prevalence due to poor intake of iron and folate despite better food access [27]. Likewise, an Ethiopian DHS analysis showed that anemia persisted among wealthier urban women, highlighting that socioeconomic status does not uniformly protect against anemia [28]. Broader SSA analyses also confirm that anemia and micronutrient deficiencies remain prevalent even among women from higher socioeconomic groups [29,30]. These findings suggest that the discrepancy between

**Table 4. Multivariable multilevel logistic regression analysis of factors associated with anemia among HIV-positive women, data from DHS 2022–2024.**

| Variables | Category | Model I AOR (95% CI) | Model II AOR (95% CI) | Model III AOR (95% CI) |
|---|---|---|---|---|
| Respondent's age | 15-24 years | 1.00 | | 1.00 |
| | 25-34 years | 1.19(0.56, 2.53) | | 1.17(0.50, 2.71) |
| | 35-49 years | 1.30(0.60, 2.84) | | 1.25(0.50, 3.09) |
| Educational status | No education | 0.75(0.31, 1.79) | | 0.68(0.24, 1.96) |
| | Primary | 0.45(0.24, 0.84)* | | 0.43(0.22, 0.81)* |
| | Secondary/Higher | 1.00 | | 1.00 |
| Marital status | Unmarried | 1.27(0.70, 2.32) | | 1.34(0.73, 2.46) |
| | Married | 1.00 | | 1.00 |
| Working status | Not working | 1.36(0.81, 2.29) | | 1.32(0.78, 2.23) |
| | Working | 1.00 | | 1.00 |
| Media exposure | No | 0.40(0.21, 0.75)* | | 0.41(0.19, 0.87)* |
| | Yes | 1.00 | | 1.00 |
| Wealth index | Poor | 0.48(0.25, 0.95)* | | 0.44(0.18, 1.09) |
| | Middle | 0.58(0.29, 1.15) | | 0.62(0.30, 1.26) |
| | Rich | 1.00 | | 1.00 |
| Currently pregnant | No | 1.00 | | 1.00 |
| | Yes | 2.20(0.39, 12.6) | | 2.21(0.38, 12.7) |
| Currently breastfeeding | No | 1.00 | | 1.00 |
| | Yes | 1.72(0.94, 3.15) | | 1.73(0.94, 3.16) |
| Terminated pregnancy | No | 1.00 | | 1.00 |
| | Yes | 0.86(0.43, 1.70) | | 0.77(0.38, 1.56) |
| Body mass index | Underweight | 1.00 | | 1.00 |
| | Normal | 0.47(0.11, 2.00) | | 0.46(0.11, 1.98) |
| | Overweight | 0.39(0.09, 1.74) | | 0.40(0.09, 1.76) |
| | Obese | 0.53(0.11, 2.45) | | 0.52(0.11, 2.40) |
| Distance to health facility | Big problem | 1.30(0.72, 2.33) | | 1.25(0.69, 2.27) |
| | Not a big problem | 1.00 | | 1.00 |
| Type of toilet facility | Unimproved | 1.18(0.68, 2.04) | | 1.23(0.70, 2.16) |
| | Improved | 1.00 | | 1.00 |
| Source of drinking water | Unimproved | 1.02(0.58, 1.82) | | 1.00(0.56, 1.78) |
| | Improved | 1.00 | | 1.00 |
| Modern contraceptive use | No | 2.42(1.38, 4.23)* | | 2.37(1.35, 4.17)* |
| | Yes | 1.00 | | 1.00 |
| Sex of the household head | Male | 1.03(0.58, 1.82) | | 1.06(0.59, 1.88) |
| | Female | 1.00 | | 1.00 |
| Health insurance coverage | No | 0.87(0.20, 3.73) | | 0.83(0.19, 3.58) |
| | Yes | 1.00 | | 1.00 |
| Menstruation within six weeks | No | 1.00 | | 1.00 |
| | Yes | 1.19(0.67, 2.10) | | 1.18(0.66, 2.10) |
| Iron supplementation | No | 2.12(1.02, 4.42)* | | 2.17(1.04, 4.55)* |
| | Yes | 1.00 | | 1.00 |
| Place of residence | Urban | | 1.00 | 1.00 |
| | Rural | | 0.95(0.74, 1.22) | 0.87(0.44, 1.73) |

*(Continued)*

**Table 4.** (Continued)

| Variables | Category | Model I AOR (95% CI) | Model II AOR (95% CI) | Model III AOR (95% CI) |
|---|---|---|---|---|
| Community literacy level | Low | | 1.03(0.79, 1.34) | 1.09(0.54, 2.16) |
| | High | | 1.00 | 1.00 |
| Community media exposure | Low | | 1.14(0.87, 1.48) | 0.97(0.49, 1.94) |
| | High | | 1.00 | 1.00 |
| Community poverty level | Low | | 1.00 | 1.00 |
| | High | | 0.78(0.60, 1.02) | 1.27(0.61, 2.64) |

*Statistically significant at p-value < 0.05.

this result and those of the 18-country study may reflect variations in country selection, survey years, or contextual factors such as urban residence, dietary diversity, and stress-related lifestyle influences. Women who had no media exposure were less likely to have anemia than their counterparts. This counterintuitive result may reflect differences in lifestyle, dietary practices, or socioeconomic conditions, as women with greater media exposure may be more influenced by urban living patterns, dietary changes, or stress factors that increase anemia risk. Similar paradoxical associations have been reported elsewhere. For example, DHS analyses across sub-Saharan Africa found that while media exposure was associated with maternal health service utilization, it was also clustered in urban populations where dietary transitions may increase anemia risk [31]. Likewise, studies in Ethiopia and Bangladesh reported that media exposure improved health knowledge and antenatal care utilization but did not consistently translate into better nutritional outcomes [32,33]. These findings suggest that women without media exposure may rely more on traditional diets and community support systems, which could provide protective effects against anemia. Overall, media exposure should be interpreted not only as a proxy for information access but also as an indicator of broader social and environmental influences on health outcomes.

The odds of anemia were higher among HIV-positive women who did not utilize modern contraceptives. This finding is in line with a study conducted in 18 SSA countries [13]. A study conducted in Ethiopia similarly reported that the use of hormonal contraceptive methods reduces anemia among women of childbearing age [34]. This might be explained by the protective role of modern contraceptive use, which can reduce menstrual blood loss, improve birth spacing, and lower the risk of nutritional depletion, thereby decreasing the likelihood of anemia among HIV-positive women. In addition, women who utilize modern contraceptives may have greater access to healthcare services and counseling, which can improve their nutritional status and overall management of HIV. Conversely, those who do not use contraceptives may experience more frequent pregnancies and higher physiological demands, further increasing their vulnerability to anemia. Non-adherence to iron supplementation during pregnancy increased the odds of developing anemia among HIV-positive women. This finding is consistent with studies conducted in Ethiopia [19] and Zimbabwe [11]. This might be due to the critical role of iron supplementation in replenishing maternal iron stores, preventing nutritional depletion, and compensating for the increased physiological demands of pregnancy. HIV-positive women are particularly vulnerable, as the infection itself can exacerbate micronutrient deficiencies and impair absorption. Failure to adhere to supplementation therefore compounds these risks, leading to a higher likelihood of anemia. These results highlight the importance of strengthening adherence support and counseling within antenatal and HIV care programs to ensure that iron supplementation is effectively utilized.

## 4.1. Strengths and limitations of the study

This study has a number of strengths, including its focus on HIV-positive women, a population often underrepresented in anemia research, and the integration of multiple sociodemographic, reproductive health, and behavioral factors, which provides a comprehensive understanding of anemia risk. The use of nationally representative data and comparison with

findings from other countries further enhances the validity and generalizability of the results. However, the study also has limitations. Its cross-sectional design restricts causal inference, and reliance on self-reported variables such as contraceptive use, media exposure, and supplementation adherence may introduce recall or social desirability bias. In addition, unmeasured confounders such as dietary diversity, micronutrient intake, and co-infections were not fully captured, and regional variations in healthcare access may limit generalizability.

## 5. Conclusions

Anemia remains a significant public health problem among HIV-positive women, influenced by factors such as contraceptive use, iron supplementation adherence, educational attainment, and media exposure. Strengthening antenatal and HIV care programs, integrating family planning services, and promoting adherence to iron supplementation are essential strategies to reduce anemia risk. Tailored health communication and nutritional interventions should also be prioritized, while future longitudinal studies are needed to establish causal relationships and address unmeasured confounders.

## Author contributions

**Conceptualization:** Enyew Getaneh Mekonen.

**Data curation:** Enyew Getaneh Mekonen.

**Formal analysis:** Enyew Getaneh Mekonen.

**Investigation:** Enyew Getaneh Mekonen.

**Methodology:** Enyew Getaneh Mekonen.

**Software:** Enyew Getaneh Mekonen.

**Supervision:** Enyew Getaneh Mekonen.

**Validation:** Enyew Getaneh Mekonen.

**Visualization:** Enyew Getaneh Mekonen.

**Writing – original draft:** Enyew Getaneh Mekonen.

**Writing – review & editing:** Enyew Getaneh Mekonen.

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
