## [Decision Letter · Decision Letter 0]

12 Jan 2026

Anemia among HIV-Positive Women in LMICs: Multilevel Analysis of Recent DHS Survey

PLOS One

Dear Dr. Mekonen,

Thank you for submitting your manuscript to PLOS ONE. After careful consideration, we feel that it has merit but does not fully meet PLOS ONE’s publication criteria as it currently stands. Therefore, we invite you to submit a revised version of the manuscript that addresses the points raised during the review process.

We look forward to receiving your revised manuscript.

Kind regards,

Mihiretu Alemayehu Arba, PhD

Academic Editor

PLOS One

Journal Requirements:

Reviewers' comments:

Reviewer's Responses to Questions

**Comments to the Author**

1. Is the manuscript technically sound, and do the data support the conclusions?

Reviewer #1: Yes

Reviewer #2: Yes

2. Has the statistical analysis been performed appropriately and rigorously?

Reviewer #1: Yes

Reviewer #2: Yes

3. Have the authors made all data underlying the findings in their manuscript fully available?

Reviewer #1: Yes

Reviewer #2: Yes

4. Is the manuscript presented in an intelligible fashion and written in standard English?

Reviewer #1: Yes

Reviewer #2: Yes

Reviewer #1: The study titled "Anemia among HIV-Positive Women in LMICs: Multilevel Analysis of Recent DHS Survey" by Mekonen addresses an important topic; however, the following points should be considered

1. Pages 8-10: The rationale for this study appears weak. While the topic of anemia among HIV-positive women in LMICs is important, there is already a substantial body of literature addressing anemia prevalence, risk factors, and outcomes in this population. The manuscript does not clearly explain what specific gap in knowledge it seeks to fill or how it adds new insights beyond existing studies. The authors should clarify what novel contribution this study provides.

2. Pages 8-10: Much of the introduction is generic and does not clearly highlight the significance of this research. The manuscript does not adequately explain why addressing anemia among HIV-positive women in LMICs is important or how this study will meaningfully advance understanding or inform interventions. Sentences such as “By leveraging large-scale, cross-country datasets, the study seeks to inform targeted interventions, support global health equity, and contribute to the evidence base” are overly broad and do not specify the concrete impact of the study. Additionally, the link between HIV and anemia is not sufficiently emphasized, and the rationale does not clearly justify why this particular population or dataset requires investigation. Strengthening these points would improve the clarity and relevance of the study’s objectives.

3. Pages 10-13: The study population is not adequately described. It is unclear whether all HIV-positive women were included regardless of ART status, pregnancy, or comorbid conditions. The manuscript should provide a more explicit description of inclusion and exclusion criteria.

4. Page 16: The authors should move the detailed definitions and formulas for ICC, MOR, and PCV from the main manuscript to the supplementary materials. Including these technical equations in the main text disrupts readability and is better suited for supplemental documentation.

5. Page 20: The manuscript attempts to compare and contrast its findings with previous studies but provides insufficient contextual information about the referenced study. For example, when stating that the finding contradicts “a study conducted in 18 SSA countries,” the authors do not summarize the study’s design, population, or key results, making it difficult for readers to understand the relevance of the comparison. The authors should provide sufficient contextual information about the cited studies to enhance interpretability.

Reviewer #2: Greetings, I am delighted to have taken a part in reviewing this paper, and below is my recommendations list:

Firstly, the paper formatting does not meet PLOS ONE requirements, the lines are not numbered, nor the headings and subheadings, please take time to review PLOS ONE style and formatting requirements and edit the paper accordingly.

Secondly, the section after abstract is known as the “Introduction” and it should contain 3 elements at least, the back ground, rational of the study, and the aims of it. Only background and study aim were present in the section.

Thirdly, as for the study rational, the discussion part included some lines at the beginning talking about the importance of this study, this should be mentioned clearly in the introduction section, along with answering the comment questions regarding what gaps in knowledge does this study cover, or what improvement in health care sector does it provide, does it promote for more research regarding the topic and in what domain, etc…. Then, you can emphasize on the rational again briefly in the discussion section.

Fourthly, the explanation regarding why HIV positive prevalence was less in our study than the Nepal study ( ref 12) is not considered a strong argument, females are known to have more prominent and even higher rates of iron deficiency anemia than men, since your study is female focused only it should still give higher prevalence than the Nepal study. However, the difference could result from more individuals screened in the Nepal study than your study, or even more severe cases of anemia identified. I recommend the authors to look more into this issue in order to generate a stronger evidence based argument.

Fifthly, the discussion part justifying that urban areas and higher socioeconomic conditions are associated with higher rates on anemia due to complex dietary influences should also be reinforced with studies that prove the same result and support the argument, the discussion must be strengthened with evidence, listing every supportive reference next to its argument.

Sixthly, the same concept for the previous point goes to the media exposure.

Seventhly, tables must be within the main text, figures must be named in main text, and referenced at the end of the submitted paper.

Lastly, for abbreviations, they must be mentioned at first appearance in text, rather than after the acknowledgment for ease of the readers.

.

Reviewer #1: No

Reviewer #2: No

---

## [Author Response · Author response to Decision Letter 1]

29 Jan 2026

Response to reviewers

Reviewer 1 Pages 8-10: The rationale for this study appears weak. While the topic of anemia among HIV-positive women in LMICs is important, there is already a substantial body of literature addressing anemia prevalence, risk factors, and outcomes in this population. The manuscript does not clearly explain what specific gap in knowledge it seeks to fill or how it adds new insights beyond existing studies. The authors should clarify what novel contribution this study provides. Thank you very much for this insightful comment. I have revised the rationale section to emphasize this contribution more clearly.

Pages 8-10: Much of the introduction is generic and does not clearly highlight the significance of this research. The manuscript does not adequately explain why addressing anemia among HIV-positive women in LMICs is important or how this study will meaningfully advance understanding or inform interventions. Sentences such as “By leveraging large-scale, cross-country datasets, the study seeks to inform targeted interventions, support global health equity, and contribute to the evidence base” are overly broad and do not specify the concrete impact of the study. Additionally, the link between HIV and anemia is not sufficiently emphasized, and the rationale does not clearly justify why this particular population or dataset requires investigation. Strengthening these points would improve the clarity and relevance of the study’s objectives.

Thank you for this constructive feedback. I have revised this to strengthen the rationale by (1) emphasizing the clinical and public health importance of anemia in HIV-positive women, (2) clarifying the link between HIV infection and anemia, and (3) explaining why the use of recent DHS datasets from nine LMICs provides a novel contribution.

Pages 10-13: The study population is not adequately described. It is unclear whether all HIV-positive women were included regardless of ART status, pregnancy, or comorbid conditions. The manuscript should provide a more explicit description of inclusion and exclusion criteria.

Thank you for this helpful comment. I provide a more explicit account of the inclusion and exclusion criteria.

Page 16: The authors should move the detailed definitions and formulas for ICC, MOR, and PCV from the main manuscript to the supplementary materials. Including these technical equations in the main text disrupts readability and is better suited for supplemental documentation.

I appreciate your suggestion regarding the placement of the detailed definitions and formulas for ICC, MOR, and PCV. While I understand the concern about readability, I believe that including these formulas in the main text strengthens methodological transparency. These measures are central to my analysis, and presenting their definitions alongside the results ensures that readers can fully understand and evaluate my approach. I trust this will balance clarity with methodological rigor.

Page 20: The manuscript attempts to compare and contrast its findings with previous studies but provides insufficient contextual information about the referenced study. For example, when stating that the finding contradicts “a study conducted in 18 SSA countries,” the authors do not summarize the study’s design, population, or key results, making it difficult for readers to understand the relevance of the comparison. The authors should provide sufficient contextual information about the cited studies to enhance interpretability.

Thank you very much for this comment. I have revised it to provide more contextual information about the referenced studies, including design, population, and key findings.

Reviewer 2 Firstly, the paper formatting does not meet PLOS ONE requirements, the lines are not numbered, nor the headings and subheadings. Please take time to review PLOS ONE style and formatting requirements and edit the paper accordingly.

Thank you very much, and I have edited it accordingly.

Secondly, the section after the abstract is known as the “Introduction,” and it should contain at least 3 elements: the background, the rationale of the study, and the aims of it. Only background and study aim were present in the section.

I agree that the introduction should contain background, rationale, and study aims. The revised section now includes all three elements.

Thirdly, as for the study rationale, the discussion part included some lines at the beginning talking about the importance of this study; this should be mentioned clearly in the introduction section, along with answering the comment questions regarding what gaps in knowledge this study covers, what improvement in the healthcare sector it provides, whether it promotes more research regarding the topic and in what domain, etc. Then, you can emphasize the rational again briefly in the discussion section.

I agree that the rationale should be clearly presented in the Introduction rather than only in the Discussion. I have revised the introduction. I also briefly re-emphasize the rationale in the discussion to reinforce the importance of these findings.

Fourthly, the explanation regarding why HIV-positive prevalence was less in our study than in the Nepal study (ref. 12) is not considered a strong argument; females are known to have more prominent and even higher rates of iron deficiency anemia than men, and since your study is female-focused only, it should still give higher prevalence than the Nepal study. However, the difference could result from more individuals screened in the Nepal study than in your study, or even more severe cases of anemia identified. I recommend the authors look more into this issue in order to generate a stronger evidence-based argument.

I agree that the initial explanation was insufficient. I have revised the discussion to provide a stronger evidence‑based argument, highlighting differences in sample size, survey years, nutritional context, and case definitions between the current study and the Nepal study. These factors may account for the observed discrepancy in anemia prevalence.

Fifthly, the discussion part justifying that urban areas and higher socioeconomic conditions are associated with higher rates of anemia due to complex dietary influences should also be reinforced with studies that prove the same result and support the argument; the discussion must be strengthened with evidence, listing every supportive reference next to its argument.

Thank you for this comment. I have strengthened the discussion by incorporating evidence from other studies that report similar paradoxical associations between higher education, urban residence, and anemia risk.

Sixthly, the same concept for the previous point goes to the media exposure. Thank you for this comment. I have revised it like the above.

Seventhly, tables must be within the main text, figures must be named in the main text, and referenced at the end of the submitted paper.

Thank you very much. I have corrected it.

Lastly, for abbreviations, they must be mentioned at the first appearance in the text, rather than after the acknowledgment, for the ease of the readers.

Thank you for this helpful observation. I have revised the manuscript so that all abbreviations are defined at their first appearance in the text.

Thank you very much for your time and effort.

---

## [Decision Letter · Decision Letter 1]

7 Apr 2026

Anemia among HIV-Positive Women in LMICs: Multilevel Analysis of Recent DHS Survey

PONE-D-25-61480R1

Dear Dr. Mekonen,

We’re pleased to inform you that your manuscript has been judged scientifically suitable for publication and will be formally accepted for publication once it meets all outstanding technical requirements.

Kind regards,

Elisabetta Pilotti

Academic Editor

PLOS One

Additional Editor Comments (optional):

Reviewers' comments:

Reviewer's Responses to Questions

**Comments to the Author**

Reviewer #1: All comments have been addressed

2. Is the manuscript technically sound, and do the data support the conclusions?

Reviewer #1: Yes

3. Has the statistical analysis been performed appropriately and rigorously?

Reviewer #1: Yes

4. Have the authors made all data underlying the findings in their manuscript fully available?

Reviewer #1: Yes

5. Is the manuscript presented in an intelligible fashion and written in standard English?

Reviewer #1: Yes

Reviewer #1: (No Response)

.

Reviewer #1: No

---

## [Editor Report · Acceptance letter]

PONE-D-25-61480R1

PLOS One

Dear Dr. Mekonen,

I'm pleased to inform you that your manuscript has been deemed suitable for publication in PLOS One. Congratulations! Your manuscript is now being handed over to our production team.

Kind regards,

on behalf of

Dr. Elisabetta Pilotti

Academic Editor

PLOS One